# Treatment with a Probiotic Mixture Containing *Bifidobacterium animalis* Subsp. *Lactis* BB12 and *Enterococcus faecium* L3 for the Prevention of Allergic Rhinitis Symptoms in Children: A Randomized Controlled Trial

**DOI:** 10.3390/nu13041315

**Published:** 2021-04-16

**Authors:** Caterina Anania, Vincenza Patrizia Di Marino, Francesca Olivero, Daniela De Canditiis, Giulia Brindisi, Federico Iannilli, Giovanna De Castro, Anna Maria Zicari, Marzia Duse

**Affiliations:** 1Immunology and Allergology Unit, Department of Mother-Child, Urological Science, Sapienza University of Rome, 00161 Rome, Italy; vincenzapatrizia.dimarino@uniroma1.it (V.P.D.M.); giulia.brindisi@uniroma1.it (G.B.); fedk4l@hotmail.it (F.I.); giovanna.decastro@uniroma1.it (G.D.C.); annamaria.zicari@uniroma1.it (A.M.Z.); marzia.duse@uniroma1.it (M.D.); 2Pediatric Clinic, Department of Pediatrics, Fondazione IRCSS Policlinico San Matteo, University of Pavia, 27100 Pavia, Italy; francesca.olivero02@universitadipavia.it; 3Institute of Applied Calculus-CNR Rome, 00185 Rome, Italy; d.decanditiis@iac.crn.it

**Keywords:** allergic rhinitis, probiotics, *Bifidobacterium animalis* subsp. *Lactis* BB12, *Enterococcus faecium L3*, children

## Abstract

Background: Probiotics may prevent the allergic response development due to their anti-inflammatory and immunomodulatory effects. The aim of this study is to determine if the prophylactic treatment with a mixture of *Bifidobacterium animalis* subsp. *Lactis* BB12 and *Enterococcus faecium* L3 would reduce symptoms and need for drug use in children with allergic rhinitis (AR). Methods: The study included 250 children aged from 6 to 17 years, affected by AR. Patients were randomly assigned to the intervention group (150) or to the placebo group (100). Patients in the intervention group, in addition to conventional therapy (local corticosteroids and/or oral antihistamines), were treated in the 3 months preceding the onset of symptoms related to the presence of the allergen to which the children were most sensitized, with a daily oral administration of a probiotic mixture containing the *Bifidobacterium animalis* subsp. *Lactis* BB12 DSM 15954 and the *Enterococcus faecium* L3 LMG P-27496 strain. We used Nasal Symptoms Score (NSS) to evaluate AR severity before and after the treatment with probiotics or placebo. Results: the patients in the intervention group had a significant reduction in their NSS after probiotic treatment (*p*-value = 2.2 × 10^−10^. Moreover, for the same group of patients, we obtained a significant reduction in the intake of pharmacological therapy. In particular, we obtained a reduction in the use of oral antihistamines (*p*-value = 2.2 × 10^−16^), local corticosteroids (*p*-value = 2.2 × 10^−13^), and of both drugs (*p*-value 1.5 × 10^−15^). Conclusions: When administered as a prophylactic treatment, a mixture of BB12 and L3 statistically decreased signs and symptoms of AR and reduced significantly the need of conventional therapy.

## 1. Introduction

Allergic diseases are increasing considerably in both adults and children and represent a public health problem, given the risk of serious complications, poor quality of life, and related costs. It is estimated that more than 20% of the population is affected by an allergic pathology such as allergic rhinitis (AR), asthma, food allergy, and/or atopic dermatitis [1]. AR affects 10 to 25% of the general population, and the prevalence of this condition has increased during the past decades, making it a global health problem [2]. The presence of specific bacterial strains could influence the development of allergic diseases [3,4]. Recently the intestinal microbiota hypothesis has been proposed to explain the rising incidence of allergic diseases [5].

AR is a non-infectious inflammatory disease of the nasal mucosa, induced by an IgE-mediated reaction, which occurs following the exposure to one or more allergens to which the patient is sensitized [6]. It is one of the most common chronic conditions in pediatric patients, and it is clinically characterized by nasal symptoms such as congestion, sneezing, itching, rhinorrhea, often associated with ocular symptoms, ear infections, and general symptoms such as asthenia and malaise [7]. Furthermore, AR is closely linked to other airway diseases like asthma, nasal polyps, sinusitis, and otitis media. Therefore, AR can significantly compromise the quality of life of children leading to poor sleep quality and lack of concentration, affecting also their growth and development [8]. Allergen immunotherapy (ITS) consists in the administration of increasing quantities of allergen in order to reduce allergic symptoms, and it is the only treatment that allows modification of the natural course of the atopic march [9]. Traditional medical treatment involves the use of local corticosteroids and oral antihistamines, which, however, do not allow the complete resolution of allergic symptoms and are often associated with side effects such as fatigue and sleepiness that limit children’s daily activities [10]. Moreover, the management of allergic diseases is expensive, both for the drug costs and for the reduction in children’s school attendance with loss of several working days of their parents [11]. Recently, it has been suggested that probiotics could modulate the immunologic and inflammatory response, and they could represent a possible preventive treatment for allergic diseases including AR. The World Health Organization (WHO) defines probiotics as “live microorganisms that, when administered in adequate amounts, are able to confer a health benefit on the host” [12]. Recent evidence supports the role of *Bifidobacteria* strains, such as *Bifidobacterium animalis* and *Bifidobacterium longum* in immune system modulation and in eliciting an anti-allergic response in early life [13]. The human gastrointestinal tract harbors 500–1000 distinct bacterial species [14]. The intestinal microbiota phyla in adults belong to six prevalent groups: *Firmicutes*, *Bacteroidetes*, *Actinobacteria*, *Proteobacteria*, *Tenericutes* and *Fusobacteria* [15]. *Actinobacteria* and specifically the genus *Bifidobacterium* are the dominant bacteria in infants [16]. In breast-fed newborns, *Bifidobacterium breve (B. breve)* is the dominant species, followed by *Bifidobacterium bifidum (B. bifidum)*, *Bifidobacterium longum* (*B. longum),* and *Bifidobacterium adolescentis* (*B. adolescentis)* [17,18]. Bacterial colonization of the intestine begins even before birth, through the ingestion by the fetus of bacteria contained in the amniotic fluid. Several environmental interactions, such as the mode of delivery, gestational age, type of breastfeeding, and nutrition, are fundamental in the development of the intestinal microbiome [19,20].

Results of the Fujimura study [21] and of the Bjorksten study [22] showed that even from the first weeks of life there are differences between the intestinal microflora of an allergic child compared to a healthy one (before the development of any clinical manifestation of atopy), and that these differences still remain at two years of age, namely, a low level of *Bifidobacteria* and *Bacteroides* and, on the contrary, an increased level of *Enterobacteria* (*E. coli*), *Staphylococcus aureus* and, only in the first three weeks, *Clostridium difficile.* The microbiome is a key element for the development of the immune response, particularly during early childhood, establishing an effective “cross-talking” with the host. Within the normal microbiome there is a balance between Gram-positive and Gram-negative bacteria. Disruption of this balance and the increase in Gram-negative bacteria (*Bacteroidetes*, *Proteobacteria*) with a consequent increase in microbial products such as lipopolysaccharide (LPS) causes alterations in the immune system leading to macrophage activation with the production of tumor necrosis factor α (TNF-α), which promotes the transition of naïve T cells to Th2 cells with a negative effect on tight junction permeability. Excess of LPS also leads to a reduction in regulatory T cells (Tregs) and therefore to an amplification of the effects of TNF-α and Th2 differentiation. The components of the normal microflora are able to induce a state of “physiological” inflammatory response in the intestine, maintained by balanced and controlled responses [23]. Probiotics help to preserve intestinal homeostasis by modulating the immune response and inducing the development of Tregs [24].

They must already be normally present in the gastrointestinal tract, are able to colonize and adhere to the epithelium, and protect the host by modulating the intestinal microbiota. This may be possible by improving the natural functions of the gastrointestinal barrier (tight junctions, mucous barrier), modulating the immune response (increase in secretory IgA, anti-inflammatory cytokines, Tregs, and NK) and antagonizing pathogens (bacteriocin production and short-chain fatty acids) [25].

*Bifidobacteria* have been shown to interact with human immune cells by modulating specific pathways, involving both innate and adaptive immunity [26]. In particular, the increase in *Bifidobacteria* leads to a decrease in Gram-negative bacteria and consequently LPS; they also strengthen the tight junctions by decreasing the permeation of LPS. BB12 in particular promotes the Th1 response, diminishing the Th2 polarization, and increases the production of secretory IgA [27]. *Enterococcus faecium* L3 is a natural enhancer of BB12: it creates the space that is occupied by the *Bifidobacteria*, increasing their growth. It also produces bacteriocins with a microbicidal action that antagonize pro-inflammatory Gram-negative bacteria (*Proteobacteria*) with a reduction in LPS-mediated subclinical inflammation, promoting an increase in IL-10 [28]. Some preliminary studies have shown that the prophylactic administration of a probiotic mixture containing *Bifidobacterium animalis* subsp. *Lactis* BB12 and *Enterococcus faecium* L3 in children affected by seasonal allergic diseases reduces the symptoms of rhinitis by about 50% [14]; it also reduces the need of oral and local corticosteroids and oral antihistamines, limiting their use and any side effects [29].

We performed a randomized controlled trial in children affected by AR, treated with local corticosteroids and oral antihistamines, to assess whether a prophylactic treatment with a probiotic mixture containing *Bifidobacterium animalis* subsp. *Lactis* BB12 and *Enterococcus faecium L3* administered before the period of allergen exposure, could reduce allergic signs and symptoms and the need of conventional therapies (local corticosteroids and oral antihistamines).

## 2. Methods

### 2.1. Participants and Design

The present study was conducted as a prospective and double-blind, randomized, placebo-controlled trial. From July 2019 to November 2020, children and adolescents affected by AR between 6 and 17 years of age were consecutively enrolled at the Department of Pediatrics, Division of Allergy and Immunology, Sapienza University of Rome. The diagnosis of AR was confirmed by pediatricians trained in allergic diseases on the basis of clinical history, skin-prick tests (SPTs), serum allergen-specific IgE towards aeroallergens, and according to Allergic Rhinitis and Its Impact on Asthma (ARIA) guidelines [30].

All the patients had a prescription with a conventional pharmacological treatment for AR such as local corticosteroids and/or oral antihistamine, and all the enrolled patients were under these treatments before and during the study.

Inclusion criteria were the following: children and adolescents aged between 6 and 17 years, attending our clinics, with a diagnosis of AR based on clinical examination, SPTs, and serum allergen-specific IgE, following ARIA guidelines [30].

Patients were already undergoing treatment with conventional AR therapies. Antihistamines (cetirizine) were taken twice per day (5 mg) for at least one month, while a local corticosteroid (beclomethasone dipropionate) was administered once a day (100 mcg, 1 puff twice per day) for 15 days per month, for three months [30]. Patients were monitored with regard to the pharmacological treatment of rhinitis through a clinical diary in which the amounts, type, frequency, and duration of drug treatment (oral antihistamine and/or local corticosteroids) was recorded.

The patients enrolled were sensitized to inhaled allergens (see Table 1), and the probiotic mixture was administered in the three months preceding the onset of symptoms related to the presence of the allergen to which the children were most sensitized.

Several children, among the enrolled patients, reported polysensitization to different pollens. However, based on the clinical history, we took into account the period in which most AR symptoms and the greatest discomfort occurred, relating to the presence of a particular allergen to which they were sensitized. Therefore, we administered the probiotic or the placebo in the three months preceding the presence of the allergen most responsible for their allergic symptoms.

Most of the patients were treated from January to March because they reported symptoms mainly related to grass pollen, whose flowering period is from May to June in our latitudes. Instead, children with symptoms related to dust mites were treated from July to September because, although this allergen is perennial, the highest concentration of dust mites is between October and January.

Exclusion criteria were as follows: patients with primary or secondary immunodeficiency, intrinsic asthma or wheezing secondary to an infectious etiology, current systemic infections, use of probiotics, prebiotics, antibiotics, and current or previous treatment with desensitizing therapy. The selected patients were randomly assigned to group A (probiotic treatment group) or to group B (placebo group) according to a computer-generated permuted-block randomization. In addition to conventional therapy, a daily oral administration of probiotics in the three months preceding the onset of symptoms, related to the presence of the allergen to which the children were most sensitized, was prescribed to group A children; on the other hand, a placebo was prescribed to group B patients. Both subjects and investigators were blind to the treatment groups. Study duration was 16 months.

Nasal Symptom Score (NSS), a validated pediatric questionnaire, was used to assess the severity of rhinitis before and after treatment with the probiotic [31,32].

The NSS is a written, four-item questionnaire, and it is the sum of the values reported by the patient (or their parents) for each of the individual items that compose it, evaluating both the severity (S) and duration (D). The rating scale is divided into four grades (0–3), where 0 indicates no symptoms, 1 indicates mild symptoms that are easily tolerated, 2 indicates symptoms that are bothersome but tolerable, and 3 indicates severe symptoms difficult to tolerate and that interfere with daily activities. The following symptoms were taken into consideration:nasal obstruction (stuffy nose),rhinorrhea (runny nose),sneezing, andnasal itching.

The total score can vary from a minimum of 0 to a maximum of 24 points. No symptoms: 0; mild rhinitis: 1 to 8; moderate rhinitis: 9 to 16; severe rhinitis: 17 to 24 (Table 2).

The NSS questionnaire was submitted to the patients before and after treatment with the probiotic mixture. NSS before treatment was recorded at the time of patient enrollment, interviewing the patients about their symptoms during the period of major exposure to the allergen to which they reported major discomfort based on their past clinical history in the previous year. NSS after treatment was recorded when the study was performed, at follow-up, referring to the same “allergic period” after patients underwent treatment with the probiotic or placebo for three months.

Considering the ability of *Bifidobacterium animalis* subsp. BB12 to modulate the immune system function promoting the physiological Th1 response as well as the capacity of *Enterococcus faecium* L3 to preserve endogenous *Bifidobacteria*, we decided to use a probiotic supplementation, which consisted of a mixture containing 2 × 10^9^ colony-forming units (CFUs) of *Bifidobacterium animalis* subsp. *Lactis* BB12 and 2 × 10^9^ of *Enterococcus faecium* L3. This probiotic also contains maltodextrin, oligofructose, and mono- and di-glycerides of fatty acids. It is a pack containing 14 sticks of 2.5 g. The product must be stored at a temperature between +2 °C and +8 °C and transported at temperatures below 25 °C for periods not exceeding 48 h. It must also be kept away from light and heat sources. This product is commercially available (Inatal ped^®^ Pharmaextracta). The placebo consisted of maltodextrin that looked and tasted the same as the probiotic mixture. The probiotic and placebo were administered by mouth, one sachet per day dissolved in a glass of water, with fasting. All patients were included in the safety analysis, and adverse events were registered.

This study was approved by the medical ethics review board of Sapienza University of Rome, Policlinico Umberto I. Patients’ parents or guardians signed a written consent form.

### 2.2. Outcomes

The primary outcome measure of this study was to evaluate whether children affected by AR, already treated with conventional therapies, would benefit from a probiotic treatment containing *Bifidobacterium animalis* subsp. *Lactis* BB12 and *Enterococcus faecium* L3, administered in the three months preceding allergenic exposure, and if the treatment could result in a reduction in their allergic signs and symptoms and need of conventional therapies (local corticosteroids and oral antihistamines).

### 2.3. Statistical Analysis

Since the sample size was large, we used the Gaussian approximation and tested the difference between the NSS before and after the treatment by the two-tailed paired t test. The *p*-values obtained for group A (probiotic) and for group B (placebo) are reported in Table 3 We also analyzed the difference between the use of drugs before (during the previous “allergic season” when they were only using drug therapy in the previous year) and after the probiotic or placebo treatment in the two groups. We counted the number of children using oral antihistamines, local corticosteroid, or both drugs before and after the treatment for both groups. Then for each type of drug, we tested the difference between the number of children taking that drug before and after the preventive probiotic treatment by using a two-tailed Fisher test. The *p*-values obtained for group A (probiotic) and for group B (placebo) are reported in Table 4 and Table 5 respectively.

## 3. Results

We enrolled 250 patients, which we randomly assigned to the intervention group (150) and to the placebo group (100). A total of 203 children completed the study, 117 into the intervention group and 86 into the placebo group. In particular, 33 subjects (22%) dropped out in the intervention group (23 discontinued intervention; 10 were lost in the follow-up), and 14 subjects (14%) dropped out in the placebo group (10 discontinued intervention; 4 lost in the follow-up).

The dropouts were due to the following reasons: poor adherence to the protocol, discontinuation of the treatment, and lost follow-up visits due to SARS-COV-2 infection. Hence, we conducted our statistical analysis on group A intervention (males 73; mean age 10.5 ± 3.1 SD) with a cardinality of 117 and on group B placebo (males 49; mean age 8.8 ± 3.5 SD) with a cardinality of 86. We showed the baseline data and characteristics of participants randomized to receive or not probiotic supplementation in Table 6. Moreover, in Table 3 we reported the percentages of sensitization among the enrolled patients. The probiotic treatment was well tolerated, and there were no clinically relevant side effects.

We applied the statistical analysis described in Section 2.3. Specifically, we applied the two-tailed paired *t* test to the NSS before and after the treatment both for group A (probiotic) and for group B (placebo). We obtained the *p*-values reported in Table 3 in particular, for the group A we obtained *p*-value = 2.2 × 10^16^, which gives a power of almost 1, then we can concluded that there was very strong statistical evidence of the difference between NSS before and after the treatment for the group A (probiotic). Changes in NSS score from baseline over the three months in the group A and group B are decribed in the boxplots (Figure 1 and Figure 2).

Regarding the drug use, for each drug (oral antihistamines, local corticosteroid, or both) we tested the difference between the number of children assuming that drug before (during the previous “allergic season” when they were only using drug therapy) and after the probiotic or placebo treatment by the Fisher test. We report the *p*-values in Table 4 and Table 5 for the intervention group (group A) and for the placebo group (group B), respectively. As it can be observed, for group A we found a statistically significant decrease in the intake of all types of drugs (oral antihistamines, local corticosteroid, or both), confirming our initial thesis.

Boxplot of the NSS before and after the treatment for group A: each box represents the measured NSS, the central mark is the median, the edges of the box are the 25th and 75th percentile, and the whiskers extend to the most extreme values.

Boxplot of the NSS before and after the treatment for group B: each box represents the measured NSS, the central mark is the median, the edges of the box are the 25th and 75th percentile, and the whiskers extend to the most extreme values.

## 4. Discussion

In this study, we obtained statistically significant results of the efficacy of the preventive probiotic treatment in reducing AR symptoms in children. When administered as a prophylactic treatment, the mixture of *Bifidobacterium animalis* subsp. *Lactis* BB12 and *Enterococcus faecium* L3 statistically decreases the signs and symptoms of AR and reduces significantly the use of drugs, including oral antihistamines and local corticosteroids. The results of our study are well in line with data of a previous trial reporting that *Bifidobacterium animalis* subsp. *Lactis* BB12 and *Enterococcus faecium* L3 strains, when administered three months before the onset of AR symptoms, statistically decreases allergic sign and symptoms [14]. Therefore, this study represents a further contribution to the literature concerning probiotics in the reduction in allergic disease symptoms [14]. Microbial exposure may direct the immune system away from allergic-type responses, but until now probiotic interventions have had limited success in the prevention and treatment of allergic disease symptoms, and the currently available evidence does not indicate that probiotic supplementation reduces the risk of developing allergic disease in children [5]. To the best of our knowledge, there have been few published double-blind, randomized, placebo-controlled trials examining the effect of probiotics in preventing AR symptoms in the pediatric population. Lin et al. in a randomized, double-blind, controlled trial demonstrated that *Lactobacillus salivarius* treatment reduces clinical symptoms and drug use among children with perennial AR [33]. A systematic review conducted by Peng et al. examined the role of probiotics in the prevention and treatment of allergic diseases. In this review, five studies analyzed the preventive role of probiotics in AR in children, and they found no effect in preventing this condition [34]. Cuello-Garcia et al. carried out a systematic review of randomized trials assessing the effects of any probiotic administered to pregnant women, breast-feeding mothers, and/or infants, and they found that supplementation with probiotics decreases the risk of atopic eczema in infants but does reduce the risk of other allergies including AR [35]. Miraglia Del Giudice et al. [36] performed a randomized, double-blind, placebo-controlled study to investigate whether *Bifidobacterium* mixture (*B. longum* BB536, *B. infantis* M-63, *B. Breve* M-16V) is effective in children with seasonal allergic rhinitis and intermittent asthma. They reported that *Bifidobacterium* mixture significantly improved symptoms of AR and quality of life (QoL). *Bifidobacteria* are frequently depleted in atopic children [21] and adults [26], and L3 promotes the preservation of endogenous gut *Bifidobacteria* in children [27]. On the basis of this background, we conducted the present study, and we found the beneficial effects of a probiotic treatment (*Bifidobacterium animalis* subsp. *Lactis* BB12 and *Enterococcus faecium* L3) in reducing AR symptoms and the use of conventional therapies in AR children. Our study presents some limitations, including the polysensitization to several pollens among the enrolled patients and the absence of the use of objective instrumental tools in the diagnostic pathway; however, this study is a further scientific contribution in a field lacking pediatric trials.

## 5. Conclusions

Several clinical trials have reported the effectiveness of probiotics in controlling symptoms and improving quality of life in patients with AR. However, most of the conducted studies have shown a significant heterogeneity regarding the type of strains, dosage, timing, and outcomes, so currently available evidence does not recommend the use of probiotics for the primary prevention of allergic diseases. The World Allergy Organization (WAO) suggests probiotic supplementation in pregnant/lactating women and in infants with a family history of allergic disease. This study demonstrates a reduced incidence of AR symptoms and a reduction in medical conventional therapies in children and adolescents previously treated with probiotic containing specific strains (*Bifidobacterium* animalis subsp. *Lactis* BB12 and *Enterococcus faecium* L3). Probiotic intervention may have a promising role in the prevention of AR, but there is still need of further studies to define the role of probiotics in preventing allergic diseases including AR. Our future goal is to confirm these results by conducting a study that also includes the use of instrumental diagnostic means, such a rhinomanometry, and studying the effects of these probiotics also in asthmatic subjects by means of spirometry.

## Figures and Tables

**Figure 1 nutrients-13-01315-f001:**
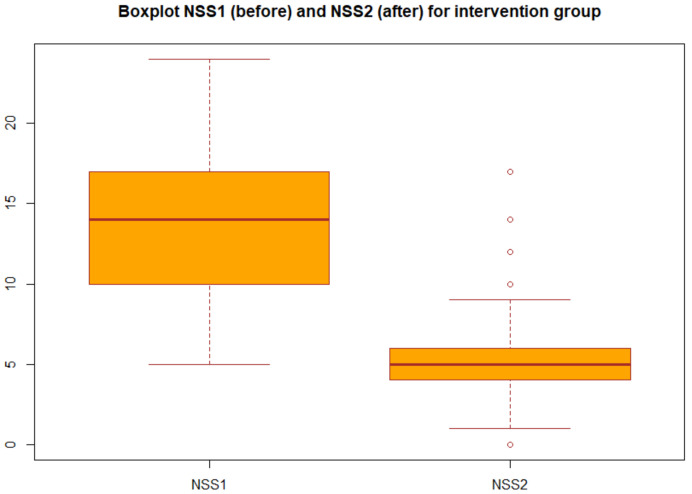
Boxplot of NSS before and after treatment in the intervention group (group A).

**Figure 2 nutrients-13-01315-f002:**
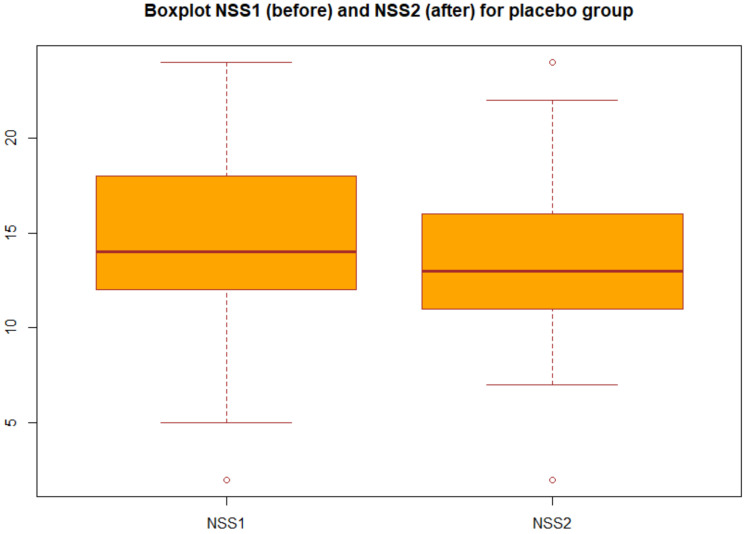
Boxplot of NSS before and after treatment in the placebo group (group B).

**Table 1 nutrients-13-01315-t001:** Prevalent allergens in the study population.

ALLERGEN	%
PARIETARIA	0.85
DPT, DPF, GRASS POLLEN	13.68
OLEA, GRASS POLLEN	0.85
DPT, DPF, PARIETARIA, GRASS POLLEN	7.70
DPT, DPF, LOLIUM, GRASS POLLEN	2.60
DPT, DPF	17.95
DPT, DPF, PARIETARIA, OLEA, CYNODON, LOLIUM, GRASS POLLEN	2.60
DPT, DPF, PARIETARIA, OLEA, GRASS POLLEN	4.27
GRASS POLLEN	5.13
LOLIUM	1.71
DPT, LOLIUM, GRASS POLLEN	3.42
DPT, DPF, ALTERNARIA	1.71
DPT, DPF, PARIETARIA, LOLIUM, GRASS POLLEN	0.85
DPT, DPF, PARIETARIA, CYNODON, LOLIUM, GRASS POLLEN	1.71
LOLIUM, GRASS POLLEN	4.27
OLEA	0.85
ALTERNARIA	0.85
CYNODON, LOLIUM, GRASS POLLEN	1.71
DPT, DPF, OLEA	3.42
OLEA, GRASS POLLEN	1.71
PARIETARIA, GRASS POLLEN	0.85
OLEA, LOLIUM, GRASS POLLEN	0.85
DPT	0.85
CYNODON, LOLIUM	1.71
ALTERNARIA, GRASS POLLEN	0.85
PARIETARIA, LOLIUM	0.85
DPT, DPF, LOLIUM	0.85
DPF, DPT, OLEA, GRASS POLLEN	3.42
DPF, DPT, ALTERNARIA, GRASS POLLEN	2.60
DPT, DPF, PARIETARIA	1.71
DPT, DPF, PARIETARIA, ALTERNARIA	0.85
DPT, DPF, OLEA, CYNODON, LOLIUM	0.85
PARIETARIA, OLEA, GRASS POLLEN	0.85
PARIETARIA, OLEA, CYNODON, LOLIUM, GRASS POLLEN	0.85
PARIETARIA, OLEA, LOLIUM, GRASS POLLEN	0.85
OLEA, CYNODON, GRASS POLLEN	0.85
LOLIUM, ALTERNARIA, GRASS POLLEN	0.85
DPF, DPT, OLEA, ALTERNARIA, GRASS POLLEN	2.60
DPT, DPF, CYNODON, LOLIUM, GRASS POLLEN	0.85
DPT, DPF, CYNODON, GRASS POLLEN	0.85

**Table 2 nutrients-13-01315-t002:** Nasal Symptom Score Scale.

Score	Clinical Significance							
0	Absence of symptoms							
1	Symptoms of mild severity							
2	Symptoms of moderate severity							
3	Symptoms of severe severity							
Score	**NASAL SYMPTOM** **Severity (S)/Duration (D)**							
	Stuffy nose S D		Runny nose S D		Sneezing S D		Nasal Itching S D	
0								
1								
2								
3								

**Table 3 nutrients-13-01315-t003:** Nasal Symptoms Score (NSS) in the study population.

	Before Treatment	After Treatment	*p* Value
**Intervention group (group A)**	14.07	5.43	2.2 × 10^16^
**Placebo group** **(group B)**	14.51	13.60	0.52

**Table 4 nutrients-13-01315-t004:** Pharmacological treatment of AR in the intervention group.

Drugs	Before Treatment	After Treatment	*p* Value
Antihistamines (oral)	67 (57%)	7 (6%)	<2.2 × 10^−16^
Corticosteroids (local)	88 (75%)	32 (27%)	2.229 × 10^−13^
Both	48 (41%)	0 (0%)	1.5 × 10^−15^

**Table 5 nutrients-13-01315-t005:** Pharmacological treatment of AR in the placebo group.

Drugs	Before Treatment	After Treatment	*p* Value
Antihistamines (oral)	66 (78%)	74 (87%)	0.16
Corticosteroids (local)	71 (84%)	51 (60%)	8.2 × 10^−04^
Both	54 (64%)	40 (40%)	0.04

**Table 6 nutrients-13-01315-t006:** Baseline characteristics of the study population.

	Probiotic (n. 117)	Placebo (n. 86)
Age (y)	10.5 ± 3.1	8.8 ± 3.5
Sex (M/F)	73/44	49/36
Rhinitis	55 (47.0%)	37 (43.02%)
Asthma + Rhinitis	50 (42.7%)	39 (45.34%)
Asthma + dermatitis	1 (0.9%)	1 (1.17%)
Rhinitis + dermatitis	6 (5.1%)	6 (6.97%)
Rhinitis + asthma + dermatitis	5 (4.3%)	3 (3.49%)
SPT_DPT	88 (75.2%)	48 (62.3%)
SPT_DPF	86 (73.5%)	47 (61.0%)
SPT_Parietary	29 (24.8%)	10 (13%)
SPT_Olea	29 (24.8%)	26 (33.8%)
SPT_Cynodon	15 (12.8%)	14 (18.2%)
SPT_Lolium	26 (22.2%)	18 (23.4%)
SPT_Grass pollen	76 (65.0%)	38 (49.4%)
SPT_Alternaria	12 (10.0%)	8 (10.4%)

SPT = Skin-prick test; DPT = *Dermatophagoides pteronissimum*; DPF = *Dermatophagoides farinae*.

## Data Availability

The data presented and analysed during the current study are available on request from the corresponding author.

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
