# Peer review of "Treatment with a Probiotic Mixture Containing Bifidobacterium animalis Subsp. Lactis BB12 and Enterococcus faecium L3 for the Prevention of Allergic Rhinitis Symptoms in Children: A Randomized Controlled Trial"

_nutrients, 2021, doi:10.3390/nu13041315_

Round 1

Reviewer 1 Report

Review of Nutrients article

I find the idea of using probiotics to treat AR very intriguing. While a few prior studies support this idea, it seems that the bulk of published data has not really found any significant improvement in AR symptoms with the use of probiotics. Therefore, to clarify this debate, the study design needs to be very precise and without flaws.

The authors seem to be using the wrong words, using “prevention” of AR when they really are intending to say “pre-seasonal” treatment. You can’t prevent an allergic disease that already has been diagnosed and treated with conventional therapies, as you stated. Therefore in line 137, I am not sure what the “..3 months preceding the onset of symptom..” really means. However, the study population is so blended with perennial and seasonal allergens that I fail to see what even “pre-seasonal” treatment would mean. It also seems that there was not a single seasonal allergen to which all study and control patients were sensitized and thus, it would be hard to interpret the “pre-seasonal” treatment. Some patients were mono-sensitized, and others were poly-sensitized. According to table 2, 18% were sensitized only to dust mites, for example. Furthermore, you have not identified what seasonal allergen you are addressing—is it trees, grass, or some other allergen? If we are discussing an increase in pollen for “season” vs. “pre-seasonal”, where are the pollen counts and how do they correlate with symptoms in both the control and study group? I believe that the study design is really just adding probiotics to patients with perennial and/or seasonal AR and determining if there is a reduction in symptom scores. While you discuss a decrease in use of pharmacological therapy, you make no attempt to quantify the amount used before and after probiotics were started. Measurement of meds used is not even defined in your methods section. Without providing me some data on amounts/class/how meds were used pre-treatment, I am not convinced that study patients had a clinically significant reduction in their use of medication.  While you provide me with general info on medication reduction for Group A, you do not report on Group B. As you know, a combination of symptom and medication scores are the accepted standard for comparing medical treatments.

The authors need to provide the intent-to-treat numbers for study/control groups, as they only report the patients that completed the study. Perhaps everyone that started completed the study actually completed the study, but that is not clear. How many dropped out and why?

I would suggest that you rework the paper and remove any reference to prevention and even pre-seasonal. Please go back to raw data and pull out the medication usage information and present detailed analysis.

Line by Line review

Line 22- I think you really mean 3 months pre-seasonal as patient already has AR

Line 64- needs rephrasing-- does the study support using probiotics for AR Tx? Ref 12 is a review article, only one ref that supports reducing AR symptoms (same as your ref # 36). Please indicate that this reviews the evidence supporting the role of these strains for anti-allergy mechanisms in early life not AR

Line 68-69:  Actinobacteria is class into which Bifidobacterium falls. So not “instead” of  but “in common” with. Indicate less microbiome diversity with the majority being Bifidobacterium.

Clarify “B. breve”,  as “Bifidobacterium breve (B. breve)” to inform the reader of what B. represents.

Line 87: do you mean overproduction as TNF alpha is important in apoptosis of triple-negative CD3/CD4/CD8 and double positive CD4/CD8 thymocytes in infants

Line 140 and following in paragraph: Paragraph is out of order should proceed the description of the two groups. See comments prior to the line by line review.

Line 162: This is confusing as 4 x 3 (max) is 12 and yet you indicate 24 as max. I presume you are adding 12-hour day and 12-hour night symptoms but you did not describe. Or did you not collect medication usage and include that in some scoring system?

Line 190: See comments in general comments

Table 3, line 201: I don't understand as I am used to seeing something less than 0.05 as being significant. Please put in the usual format. What were the medication usage scores or tabulation? I need to know what meds both groups were taking before and during study. Were groups evenly matched in terms of medication use before the study started? Do you mean that 56.41% used no meds? Perhaps the same % used no meds before study entry and the remainder had a decrease. Perhaps some had to increase meds. I cannot use the limited information you provided for meds to draw any conclusions. Of the 43% that used meds, how many used corticosteroids, how many antihistamines, how many used both? You did not indicate what happened to the medication usage in group B.  I suggest you provide # of patients vs. %. Suggest you go back and summarize the meds data and present in paper.

Line 223: Sentence ending with “and symptoms.” Provide references. Are you saying that these were SAR studies, as 3 months before the development of AR? Also correct sentence to say before onset of specific, e.g., grass season.

Line 224: Your study did not prevent allergic disease but symptom reduction in AR patients. The Di Pierro study is poorly designed, assuming that all children with AR will be worse in April-June than in Jan-March. No sensitization/seasonal symptom hx prior to study provided. All patients had atopy in this retrospective study-- "Prevention" of allergy was not truly the case.

Line 232: Please add that this (ref 33) was for perennial AR.

Line 246: Change “preventing” to “reducing”

Review of Nutrients article

I find the idea of using probiotics to treat AR very intriguing. While a few prior studies support this idea, it seems that the bulk of published data has not really found any significant improvement in AR symptoms with the use of probiotics. Therefore, to clarify this debate, the study design needs to be very precise and without flaws.

The authors seem to be using the wrong words, using “prevention” of AR when they really are intending to say “pre-seasonal” treatment. You can’t prevent an allergic disease that already has been diagnosed and treated with conventional therapies, as you stated. Therefore in line 137, I am not sure what the “..3 months preceding the onset of symptom..” really means. However, the study population is so blended with perennial and seasonal allergens that I fail to see what even “pre-seasonal” treatment would mean. It also seems that there was not a single seasonal allergen to which all study and control patients were sensitized and thus, it would be hard to interpret the “pre-seasonal” treatment. Some patients were mono-sensitized, and others were poly-sensitized. According to table 2, 18% were sensitized only to dust mites, for example. Furthermore, you have not identified what seasonal allergen you are addressing—is it trees, grass, or some other allergen? If we are discussing an increase in pollen for “season” vs. “pre-seasonal”, where are the pollen counts and how do they correlate with symptoms in both the control and study group? I believe that the study design is really just adding probiotics to patients with perennial and/or seasonal AR and determining if there is a reduction in symptom scores. While you discuss a decrease in use of pharmacological therapy, you make no attempt to quantify the amount used before and after probiotics were started. Measurement of meds used is not even defined in your methods section. Without providing me some data on amounts/class/how meds were used pre-treatment, I am not convinced that study patients had a clinically significant reduction in their use of medication.  While you provide me with general info on medication reduction for Group A, you do not report on Group B. As you know, a combination of symptom and medication scores are the accepted standard for comparing medical treatments.

The authors need to provide the intent-to-treat numbers for study/control groups, as they only report the patients that completed the study. Perhaps everyone that started completed the study actually completed the study, but that is not clear. How many dropped out and why?

I would suggest that you rework the paper and remove any reference to prevention and even pre-seasonal. Please go back to raw data and pull out the medication usage information and present detailed analysis.

Line by Line review

Line 22- I think you really mean 3 months pre-seasonal as patient already has AR

Line 64- needs rephrasing-- does the study support using probiotics for AR Tx? Ref 12 is a review article, only one ref that supports reducing AR symptoms (same as your ref # 36). Please indicate that this reviews the evidence supporting the role of these strains for anti-allergy mechanisms in early life not AR

Line 68-69:  Actinobacteria is class into which Bifidobacterium falls. So not “instead” of  but “in common” with. Indicate less microbiome diversity with the majority being Bifidobacterium.

Clarify “B. breve”,  as “Bifidobacterium breve (B. breve)” to inform the reader of what B. represents.

Line 87: do you mean overproduction as TNF alpha is important in apoptosis of triple-negative CD3/CD4/CD8 and double positive CD4/CD8 thymocytes in infants

Line 140 and following in paragraph: Paragraph is out of order should proceed the description of the two groups. See comments prior to the line by line review.

Line 162: This is confusing as 4 x 3 (max) is 12 and yet you indicate 24 as max. I presume you are adding 12-hour day and 12-hour night symptoms but you did not describe. Or did you not collect medication usage and include that in some scoring system?

Line 190: See comments in general comments

Table 3, line 201: I don't understand as I am used to seeing something less than 0.05 as being significant. Please put in the usual format. What were the medication usage scores or tabulation? I need to know what meds both groups were taking before and during study. Were groups evenly matched in terms of medication use before the study started? Do you mean that 56.41% used no meds? Perhaps the same % used no meds before study entry and the remainder had a decrease. Perhaps some had to increase meds. I cannot use the limited information you provided for meds to draw any conclusions. Of the 43% that used meds, how many used corticosteroids, how many antihistamines, how many used both? You did not indicate what happened to the medication usage in group B.  I suggest you provide # of patients vs. %. Suggest you go back and summarize the meds data and present in paper.

Line 223: Sentence ending with “and symptoms.” Provide references. Are you saying that these were SAR studies, as 3 months before the development of AR? Also correct sentence to say before onset of specific, e.g., grass season.

Line 224: Your study did not prevent allergic disease but symptom reduction in AR patients. The Di Pierro study is poorly designed, assuming that all children with AR will be worse in April-June than in Jan-March. No sensitization/seasonal symptom hx prior to study provided. All patients had atopy in this retrospective study-- "Prevention" of allergy was not truly the case.

Line 232: Please add that this (ref 33) was for perennial AR.

Line 246: Change “preventing” to “reducing”

Review of Nutrients article

I find the idea of using probiotics to treat AR very intriguing. While a few prior studies support this idea, it seems that the bulk of published data has not really found any significant improvement in AR symptoms with the use of probiotics. Therefore, to clarify this debate, the study design needs to be very precise and without flaws.

The authors seem to be using the wrong words, using “prevention” of AR when they really are intending to say “pre-seasonal” treatment. You can’t prevent an allergic disease that already has been diagnosed and treated with conventional therapies, as you stated. Therefore in line 137, I am not sure what the “..3 months preceding the onset of symptom..” really means. However, the study population is so blended with perennial and seasonal allergens that I fail to see what even “pre-seasonal” treatment would mean. It also seems that there was not a single seasonal allergen to which all study and control patients were sensitized and thus, it would be hard to interpret the “pre-seasonal” treatment. Some patients were mono-sensitized, and others were poly-sensitized. According to table 2, 18% were sensitized only to dust mites, for example. Furthermore, you have not identified what seasonal allergen you are addressing—is it trees, grass, or some other allergen? If we are discussing an increase in pollen for “season” vs. “pre-seasonal”, where are the pollen counts and how do they correlate with symptoms in both the control and study group? I believe that the study design is really just adding probiotics to patients with perennial and/or seasonal AR and determining if there is a reduction in symptom scores. While you discuss a decrease in use of pharmacological therapy, you make no attempt to quantify the amount used before and after probiotics were started. Measurement of meds used is not even defined in your methods section. Without providing me some data on amounts/class/how meds were used pre-treatment, I am not convinced that study patients had a clinically significant reduction in their use of medication.  While you provide me with general info on medication reduction for Group A, you do not report on Group B. As you know, a combination of symptom and medication scores are the accepted standard for comparing medical treatments.

The authors need to provide the intent-to-treat numbers for study/control groups, as they only report the patients that completed the study. Perhaps everyone that started completed the study actually completed the study, but that is not clear. How many dropped out and why?

I would suggest that you rework the paper and remove any reference to prevention and even pre-seasonal. Please go back to raw data and pull out the medication usage information and present detailed analysis.

Line by Line review

Line 22- I think you really mean 3 months pre-seasonal as patient already has AR

Line 64- needs rephrasing-- does the study support using probiotics for AR Tx? Ref 12 is a review article, only one ref that supports reducing AR symptoms (same as your ref # 36). Please indicate that this reviews the evidence supporting the role of these strains for anti-allergy mechanisms in early life not AR

Line 68-69:  Actinobacteria is class into which Bifidobacterium falls. So not “instead” of  but “in common” with. Indicate less microbiome diversity with the majority being Bifidobacterium.

Clarify “B. breve”,  as “Bifidobacterium breve (B. breve)” to inform the reader of what B. represents.

Line 87: do you mean overproduction as TNF alpha is important in apoptosis of triple-negative CD3/CD4/CD8 and double positive CD4/CD8 thymocytes in infants

Line 140 and following in paragraph: Paragraph is out of order should proceed the description of the two groups. See comments prior to the line by line review.

Line 162: This is confusing as 4 x 3 (max) is 12 and yet you indicate 24 as max. I presume you are adding 12-hour day and 12-hour night symptoms but you did not describe. Or did you not collect medication usage and include that in some scoring system?

Line 190: See comments in general comments

Table 3, line 201: I don't understand as I am used to seeing something less than 0.05 as being significant. Please put in the usual format. What were the medication usage scores or tabulation? I need to know what meds both groups were taking before and during study. Were groups evenly matched in terms of medication use before the study started? Do you mean that 56.41% used no meds? Perhaps the same % used no meds before study entry and the remainder had a decrease. Perhaps some had to increase meds. I cannot use the limited information you provided for meds to draw any conclusions. Of the 43% that used meds, how many used corticosteroids, how many antihistamines, how many used both? You did not indicate what happened to the medication usage in group B.  I suggest you provide # of patients vs. %. Suggest you go back and summarize the meds data and present in paper.

Line 223: Sentence ending with “and symptoms.” Provide references. Are you saying that these were SAR studies, as 3 months before the development of AR? Also correct sentence to say before onset of specific, e.g., grass season.

Line 224: Your study did not prevent allergic disease but symptom reduction in AR patients. The Di Pierro study is poorly designed, assuming that all children with AR will be worse in April-June than in Jan-March. No sensitization/seasonal symptom hx prior to study provided. All patients had atopy in this retrospective study-- "Prevention" of allergy was not truly the case.

Line 232: Please add that this (ref 33) was for perennial AR.

Line 246: Change “preventing” to “reducing”

Review of Nutrients article

General comments: 

I find the idea of using probiotics to treat AR very intriguing. While a few prior studies support this idea, it seems that the bulk of published data has not really found any significant improvement in AR symptoms with the use of probiotics. Therefore, to clarify this debate, the study design needs to be very precise and without flaws.

The authors seem to be using the wrong words, using “prevention” of AR when they really are intending to say “pre-seasonal” treatment. You can’t prevent an allergic disease that already has been diagnosed and treated with conventional therapies, as you stated. Therefore in line 137, I am not sure what the “..3 months preceding the onset of symptom..” really means. However, the study population is so blended with perennial and seasonal allergens that I fail to see what even “pre-seasonal” treatment would mean. It also seems that there was not a single seasonal allergen to which all study and control patients were sensitized and thus, it would be hard to interpret the “pre-seasonal” treatment. Some patients were mono-sensitized, and others were poly-sensitized. According to table 2, 18% were sensitized only to dust mites, for example. Furthermore, you have not identified what seasonal allergen you are addressing—is it trees, grass, or some other allergen? If we are discussing an increase in pollen for “season” vs. “pre-seasonal”, where are the pollen counts and how do they correlate with symptoms in both the control and study group? I believe that the study design is really just adding probiotics to patients with perennial and/or seasonal AR and determining if there is a reduction in symptom scores. While you discuss a decrease in use of pharmacological therapy, you make no attempt to quantify the amount used before and after probiotics were started. Measurement of meds used is not even defined in your methods section. Without providing me some data on amounts/class/how meds were used pre-treatment, I am not convinced that study patients had a clinically significant reduction in their use of medication.  While you provide me with general info on medication reduction for Group A, you do not report on Group B. As you know, a combination of symptom and medication scores are the accepted standard for comparing medical treatments.

The authors need to provide the intent-to-treat numbers for study/control groups, as they only report the patients that completed the study. Perhaps everyone that started completed the study actually completed the study, but that is not clear. How many dropped out and why?

I would suggest that you rework the paper and remove any reference to prevention and even pre-seasonal. Please go back to raw data and pull out the medication usage information and present detailed analysis.

Line by Line review

Line 22- I think you really mean 3 months pre-seasonal as patient already has AR

Line 64- needs rephrasing-- does the study support using probiotics for AR Tx? Ref 12 is a review article, only one ref that supports reducing AR symptoms (same as your ref # 36). Please indicate that this reviews the evidence supporting the role of these strains for anti-allergy mechanisms in early life not AR

Line 68-69:  Actinobacteria is class into which Bifidobacterium falls. So not “instead” of  but “in common” with. Indicate less microbiome diversity with the majority being Bifidobacterium.

Clarify “B. breve”,  as “Bifidobacterium breve (B. breve)” to inform the reader of what B. represents.

Line 87: do you mean overproduction as TNF alpha is important in apoptosis of triple-negative CD3/CD4/CD8 and double positive CD4/CD8 thymocytes in infants

Line 140 and following in paragraph: Paragraph is out of order should proceed the description of the two groups. See comments prior to the line by line review.

Line 162: This is confusing as 4 x 3 (max) is 12 and yet you indicate 24 as max. I presume you are adding 12-hour day and 12-hour night symptoms but you did not describe. Or did you not collect medication usage and include that in some scoring system?

Line 190: See comments in general comments

Table 3, line 201: I don't understand as I am used to seeing something less than 0.05 as being significant. Please put in the usual format. What were the medication usage scores or tabulation? I need to know what meds both groups were taking before and during study. Were groups evenly matched in terms of medication use before the study started? Do you mean that 56.41% used no meds? Perhaps the same % used no meds before study entry and the remainder had a decrease. Perhaps some had to increase meds. I cannot use the limited information you provided for meds to draw any conclusions. Of the 43% that used meds, how many used corticosteroids, how many antihistamines, how many used both? You did not indicate what happened to the medication usage in group B.  I suggest you provide # of patients vs. %. Suggest you go back and summarize the meds data and present in paper.

Line 223: Sentence ending with “and symptoms.” Provide references. Are you saying that these were SAR studies, as 3 months before the development of AR? Also correct sentence to say before onset of specific, e.g., grass season.

Line 224: Your study did not prevent allergic disease but symptom reduction in AR patients. The Di Pierro study is poorly designed, assuming that all children with AR will be worse in April-June than in Jan-March. No sensitization/seasonal symptom hx prior to study provided. All patients had atopy in this retrospective study-- "Prevention" of allergy was not truly the case.

Line 232: Please add that this (ref 33) was for perennial AR.

Line 246: Change “preventing” to “reducing”

Review of Nutrients article

I find the idea of using probiotics to treat AR very intriguing. While a few prior studies support this idea, it seems that the bulk of published data has not really found any significant improvement in AR symptoms with the use of probiotics. Therefore, to clarify this debate, the study design needs to be very precise and without flaws.

The authors seem to be using the wrong words, using “prevention” of AR when they really are intending to say “pre-seasonal” treatment. You can’t prevent an allergic disease that already has been diagnosed and treated with conventional therapies, as you stated. Therefore in line 137, I am not sure what the “..3 months preceding the onset of symptom..” really means. However, the study population is so blended with perennial and seasonal allergens that I fail to see what even “pre-seasonal” treatment would mean. It also seems that there was not a single seasonal allergen to which all study and control patients were sensitized and thus, it would be hard to interpret the “pre-seasonal” treatment. Some patients were mono-sensitized, and others were poly-sensitized. According to table 2, 18% were sensitized only to dust mites, for example. Furthermore, you have not identified what seasonal allergen you are addressing—is it trees, grass, or some other allergen? If we are discussing an increase in pollen for “season” vs. “pre-seasonal”, where are the pollen counts and how do they correlate with symptoms in both the control and study group? I believe that the study design is really just adding probiotics to patients with perennial and/or seasonal AR and determining if there is a reduction in symptom scores. While you discuss a decrease in use of pharmacological therapy, you make no attempt to quantify the amount used before and after probiotics were started. Measurement of meds used is not even defined in your methods section. Without providing me some data on amounts/class/how meds were used pre-treatment, I am not convinced that study patients had a clinically significant reduction in their use of medication.  While you provide me with general info on medication reduction for Group A, you do not report on Group B. As you know, a combination of symptom and medication scores are the accepted standard for comparing medical treatments.

The authors need to provide the intent-to-treat numbers for study/control groups, as they only report the patients that completed the study. Perhaps everyone that started completed the study actually completed the study, but that is not clear. How many dropped out and why?

I would suggest that you rework the paper and remove any reference to prevention and even pre-seasonal. Please go back to raw data and pull out the medication usage information and present detailed analysis.

Line by Line review

Line 22- I think you really mean 3 months pre-seasonal as patient already has AR

Line 64- needs rephrasing-- does the study support using probiotics for AR Tx? Ref 12 is a review article, only one ref that supports reducing AR symptoms (same as your ref # 36). Please indicate that this reviews the evidence supporting the role of these strains for anti-allergy mechanisms in early life not AR

Line 68-69:  Actinobacteria is class into which Bifidobacterium falls. So not “instead” of  but “in common” with. Indicate less microbiome diversity with the majority being Bifidobacterium.

Clarify “B. breve”,  as “Bifidobacterium breve (B. breve)” to inform the reader of what B. represents.

Line 87: do you mean overproduction as TNF alpha is important in apoptosis of triple-negative CD3/CD4/CD8 and double positive CD4/CD8 thymocytes in infants

Line 140 and following in paragraph: Paragraph is out of order should proceed the description of the two groups. See comments prior to the line by line review.

Line 162: This is confusing as 4 x 3 (max) is 12 and yet you indicate 24 as max. I presume you are adding 12-hour day and 12-hour night symptoms but you did not describe. Or did you not collect medication usage and include that in some scoring system?

Line 190: See comments in general comments

Table 3, line 201: I don't understand as I am used to seeing something less than 0.05 as being significant. Please put in the usual format. What were the medication usage scores or tabulation? I need to know what meds both groups were taking before and during study. Were groups evenly matched in terms of medication use before the study started? Do you mean that 56.41% used no meds? Perhaps the same % used no meds before study entry and the remainder had a decrease. Perhaps some had to increase meds. I cannot use the limited information you provided for meds to draw any conclusions. Of the 43% that used meds, how many used corticosteroids, how many antihistamines, how many used both? You did not indicate what happened to the medication usage in group B.  I suggest you provide # of patients vs. %. Suggest you go back and summarize the meds data and present in paper.

Line 223: Sentence ending with “and symptoms.” Provide references. Are you saying that these were SAR studies, as 3 months before the development of AR? Also correct sentence to say before onset of specific, e.g., grass season.

Line 224: Your study did not prevent allergic disease but symptom reduction in AR patients. The Di Pierro study is poorly designed, assuming that all children with AR will be worse in April-June than in Jan-March. No sensitization/seasonal symptom hx prior to study provided. All patients had atopy in this retrospective study-- "Prevention" of allergy was not truly the case.

Line 232: Please add that this (ref 33) was for perennial AR.

Line 246: Change “preventing” to “reducing”

Review of Nutrients article

I find the idea of using probiotics to treat AR very intriguing. While a few prior studies support this idea, it seems that the bulk of published data has not really found any significant improvement in AR symptoms with the use of probiotics. Therefore, to clarify this debate, the study design needs to be very precise and without flaws.

The authors seem to be using the wrong words, using “prevention” of AR when they really are intending to say “pre-seasonal” treatment. You can’t prevent an allergic disease that already has been diagnosed and treated with conventional therapies, as you stated. Therefore in line 137, I am not sure what the “..3 months preceding the onset of symptom..” really means. However, the study population is so blended with perennial and seasonal allergens that I fail to see what even “pre-seasonal” treatment would mean. It also seems that there was not a single seasonal allergen to which all study and control patients were sensitized and thus, it would be hard to interpret the “pre-seasonal” treatment. Some patients were mono-sensitized, and others were poly-sensitized. According to table 2, 18% were sensitized only to dust mites, for example. Furthermore, you have not identified what seasonal allergen you are addressing—is it trees, grass, or some other allergen? If we are discussing an increase in pollen for “season” vs. “pre-seasonal”, where are the pollen counts and how do they correlate with symptoms in both the control and study group? I believe that the study design is really just adding probiotics to patients with perennial and/or seasonal AR and determining if there is a reduction in symptom scores. While you discuss a decrease in use of pharmacological therapy, you make no attempt to quantify the amount used before and after probiotics were started. Measurement of meds used is not even defined in your methods section. Without providing me some data on amounts/class/how meds were used pre-treatment, I am not convinced that study patients had a clinically significant reduction in their use of medication.  While you provide me with general info on medication reduction for Group A, you do not report on Group B. As you know, a combination of symptom and medication scores are the accepted standard for comparing medical treatments.

The authors need to provide the intent-to-treat numbers for study/control groups, as they only report the patients that completed the study. Perhaps everyone that started completed the study actually completed the study, but that is not clear. How many dropped out and why?

I would suggest that you rework the paper and remove any reference to prevention and even pre-seasonal. Please go back to raw data and pull out the medication usage information and present detailed analysis.

Line by Line review

Line 22- I think you really mean 3 months pre-seasonal as patient already has AR

Line 64- needs rephrasing-- does the study support using probiotics for AR Tx? Ref 12 is a review article, only one ref that supports reducing AR symptoms (same as your ref # 36). Please indicate that this reviews the evidence supporting the role of these strains for anti-allergy mechanisms in early life not AR

Line 68-69:  Actinobacteria is class into which Bifidobacterium falls. So not “instead” of  but “in common” with. Indicate less microbiome diversity with the majority being Bifidobacterium.

Clarify “B. breve”,  as “Bifidobacterium breve (B. breve)” to inform the reader of what B. represents.

Line 87: do you mean overproduction as TNF alpha is important in apoptosis of triple-negative CD3/CD4/CD8 and double positive CD4/CD8 thymocytes in infants

Line 140 and following in paragraph: Paragraph is out of order should proceed the description of the two groups. See comments prior to the line by line review.

Line 162: This is confusing as 4 x 3 (max) is 12 and yet you indicate 24 as max. I presume you are adding 12-hour day and 12-hour night symptoms but you did not describe. Or did you not collect medication usage and include that in some scoring system?

Line 190: See comments in general comments

Table 3, line 201: I don't understand as I am used to seeing something less than 0.05 as being significant. Please put in the usual format. What were the medication usage scores or tabulation? I need to know what meds both groups were taking before and during study. Were groups evenly matched in terms of medication use before the study started? Do you mean that 56.41% used no meds? Perhaps the same % used no meds before study entry and the remainder had a decrease. Perhaps some had to increase meds. I cannot use the limited information you provided for meds to draw any conclusions. Of the 43% that used meds, how many used corticosteroids, how many antihistamines, how many used both? You did not indicate what happened to the medication usage in group B.  I suggest you provide # of patients vs. %. Suggest you go back and summarize the meds data and present in paper.

Line 223: Sentence ending with “and symptoms.” Provide references. Are you saying that these were SAR studies, as 3 months before the development of AR? Also correct sentence to say before onset of specific, e.g., grass season.

Line 224: Your study did not prevent allergic disease but symptom reduction in AR patients. The Di Pierro study is poorly designed, assuming that all children with AR will be worse in April-June than in Jan-March. No sensitization/seasonal symptom hx prior to study provided. All patients had atopy in this retrospective study-- "Prevention" of allergy was not truly the case.

Line 232: Please add that this (ref 33) was for perennial AR.

Line 246: Change “preventing” to “reducing”

Reviewer 2 Report

Nutrients: Treatment with a Probiotic Mixture Containing Bifidobacterium Animals Subsp. Lactis BB12 and Enterococcus Facecium L3 for the Prevention of Allergic Rhinitis in Children: A Randomized Control Trial

This article investigates the use of a probiotic mixture Bifidobacterium spp., for treatment of allergic rhinitis (AR) in pediatric participants. While this article certainly fills a gap in the research, namely, probiotic trials in pediatric populations, there are significant flaws to the study design

Major:

  • Treatment occurs in the 3 months preceding the development of AR symptoms. What allergy season was this performed? How is the start of treatment decided? Clearly, your population is heterogeneous for several allergies as shown in Table 2. How are differing allergy seasons (seasonal and perennial) controlled for in this study with regards to symptomatic data?
  • Primary outcome of a NSS is sufficient, but the authors should consider an objective measurement to support this symptomatic measure, such as PNIF or spirometry data.
  • Similarly, how is the use of medications such as local corticosteroids and/or antihistamines monitored? A diary tracker? What are the statistical tests performed on this data? This is important to ensure their symptoms are well controlled.
  • How is the probiotic administered?
  • The box plot has no figure legends and missing titles or the X and Y axis. No significance values are reported on the graphs.

Minor:

  • The definition of probiotics (WHO) should come before any mention of probiotics in the introduction.
  • NSS description should be configured as a table.
  • Line 207: NNS is a typo.
  • There seems to be several outliers in the datapoints. How can you account for these?
  • No mention of future directions into the mechanisms of action.

Round 2

Reviewer 1 Report

I believe that the authors have addressed most of my concerns, although they did not do a point by point response as they did for reviewer # 2. 

While there is still some significant flaws in study design, I believe that the reader can now understand the methods used and draw their own conclusions. 

I do not believe that further revisions are needed.

Author Response

Dear Colleague,

Thank you for your precious comments and advice.

Best Regards

Caterina Anania

Reviewer 2 Report

  • It is still unclear as to when the NSS was measured before and after treatment.  
  • The authors mention that “Lactis BB12 and Enterococcus faecium L3 in children affected by seasonal allergic diseases, reduces the rhinitis’ symptoms of rhinitis by about 50%; it also reduces the need of oral and local cortisones corticosteroids and antihistamines, limiting their use and any side effects.” There is no appropriate citation linked to this statement. Is this unpublished data? What is the justification for this probiotic? Is it commercially available?  
  • How did the authors determine which allergen was most responsible for allergic symptoms (e.g., clinical history, IgE levels)? Likewise, the authors should address the limitations of this model/design in their discussion. 
  • How was drug use analyzed? Were trends in drug use analyzed? Are there correlations between symptom data and treatment usage?    
